# Leishmania amazonensis promastigotes in 3D Collagen I culture: an *in vitro* physiological environment for the study of extracellular matrix and host cell interactions

Debora B. Petropolis[1,2], Juliany C.F. Rodrigues[1,3,4,5], Nathan B. Viana[6,7], Bruno Pontes[6], Camila F.A. Pereira[1] and Fernando C. Silva-Filho[1,8]

[1] Instituto de Biofísica Carlos Chagas Filho, Universidade Federal do Rio de Janeiro, Rio de Janeiro, Brazil
[2] Institut Pasteur, Paris, France
[3] Núcleo Multidisciplinar de Pesquisa UFRJ-Xerém (NUMPEX-BIO), Polo Avançado de Xerém, Universidade Federal do Rio de Janeiro, Brazil
[4] Instituto Nacional de Ciência e Tecnologia de Biologia Estrutural e Bioimagem, Brazil
[5] Instituto Nacional de Metrologia, Qualidade e Tecnologia, Inmetro, Brazil
[6] LPO-COPEA, Instituto de Ciências Biomédicas, Universidade Federal do Rio de Janeiro, Brazil
[7] Instituto de Física, Universidade Federal do Rio de Janeiro, Rio de Janeiro, Brazil
[8] Universidade Estadual do Norte Fluminense, Campos, Brazil

Corresponding author
Debora B. Petropolis,
dpetropo@pasteur.fr

## ABSTRACT

*Leishmania amazonensis* is the causative agent of American cutaneous leishmaniasis, an important neglected tropical disease. Once *Leishmania amazonensis* is inoculated into the human host, promastigotes are exposed to the extracellular matrix (ECM) of the dermis. However, little is known about the interaction between the ECM and *Leishmania* promastigotes. In this study we established *L. amazonensis* promastigote culture in a three-dimensional (3D) environment mainly composed of Collagen I (COL I). This 3D culture recreates *in vitro* some aspects of the human host infection site, enabling the study of the interaction mechanisms of *L. amazonensis* with the host ECM. Promastigotes exhibited "freeze and run" migration in the 3D COL I matrix, which is completely different from the conventional *in vitro* swimming mode of migration. Moreover, *L. amazonensis* promastigotes were able to invade, migrate inside, and remodel the 3D COL I matrix. Promastigote trans-matrix invasion and the freeze and run migration mode were also observed when macrophages were present in the matrix. At least two classes of proteases, metallo- and cysteine proteases, are involved in the 3D COL I matrix degradation caused by *Leishmania*. Treatment with a mixture of protease inhibitors significantly reduced promastigote invasion and migration through this matrix. Together our results demonstrate that *L. amazonensis* promastigotes release proteases and actively remodel their 3D environment, facilitating their migration. This raises the possibility that promastigotes actively interact with their 3D environment during the search for their cellular "home"—macrophages. Supporting this hypothesis, promastigotes migrated faster than macrophages in a novel 3D co-culture model.

## INTRODUCTION

The promastigote form of *Leishmania amazonensis*, causative agent of American cutaneous leishmaniasis (*Barral et al., 1991*; *Murray et al., 2005*), is transmitted through the bite of infected sand flies from the genus *Lutzomyia* (*Kaye & Scott, 2011*; *Neuber, 2008*). Once inoculated, the promastigotes are exposed to the dermis microenvironment, which is composed of extracellular matrix (ECM) proteins organized in a fibrilar network (*Rogers, 2012*). Because *L. amazonensis* is an intracellular parasite that only proliferates inside a cellular host in the mammalian host, its interaction with the host ECM has been neglected. However, in order to establish an intracellular infection, the promastigote form must overcome the obstacles presented by the dermis ECM (*Chang & McGwire, 2002*). Except for broad ECM alterations during experimental *Leishmania* infection (*Abreu-Silva et al., 2004*; *Melo et al., 2009*; *Silva-Almeida et al., 2012a*) little is known about the interaction of *L. amazonensis* with the host ECM (*Kulkarni et al., 2008*; *McGwire, Chang & Engman, 2003*). Although long viewed only as a supportive structure, the ECM is an essential part of the cell's milieu that, through direct or indirect means, regulates almost all cellular behavior (*Hynes, 2009*), including inflammatory signaling (*Larsen et al., 2006*).

The glycoprotein gp63 present on parasite surfaces is a broad-spectrum zinc-dependent metalloprotease that works as an important virulence factor during *Leishmania* infection (*McMaster et al., 1994*; *Silva-Almeida et al., 2012a*; *Yao et al., 2002*). gp63 can degrade ECM components such as fibronectin and collagen IV and seems to enhance *L. amazonensis* migration in Matrigel, a commercial 3D gelatin of basement membrane ECM proteins (*Kulkarni et al., 2008*; *McGwire, Chang & Engman, 2003*). However, collagen I (COL I) is the main component of the dermis ECM, and for the moment little is known about the interaction of *L. amazonensis* with COL I rich environments (*Lira, Rosales-Encina & Arguello, 1997*).

Here, we introduce a new model for culture of *L. amazonensis* promastigotes in an *in vitro* 3D COL I matrix, which represents a more physiological *in vitro* culture because it better mimics many extracellular aspects of the *Leishmania* environment in the human host. The objective of this work was to characterize *L. amazonensis* invasion, migration, and matrix remodeling in 3D COL I matrices, focusing on the functions of *Leishmania* proteases. The establishment of this model enables evaluation of the mechanisms of *Leishmania* interaction with host cells in a 3D environment.

## METHODS

### Ethics statement

The use of animal models was approved by the Ethics Committee for Animal Experimentation of the Health Sciences Centre, Federal University of Rio de Janeiro (Protocols n. IBCCF 096/097), according to the Brazilian federal law. All animals received humane

care in compliance with the "Principles of Laboratory Animal Care" formulated by the National Society for Medical Research and the "Guide for the Care and Use of Laboratory Animals" prepared by the National Academy of Sciences, USA.

### *Leishmania amazonensis* promastigotes culture

The MHOM/BR/75/Josefa strain of *L. amazonensis* used in this study is an anonymized strain that was isolated in 1975 from a patient with diffuse cutaneous leishmaniasis by Dr. Cesar A. Cuba-Cuba (Brasilia University, Brazil) and kindly provided by the *Leishmania* Collection of the Instituto Oswaldo Cruz (Code IOCL 0071–FIOCRUZ). It has been maintained by BALB/c footpad inoculation. Amastigote forms were obtained from these mice, and transformed into promastigotes that were axenically cultured in Warren's medium (brain and heart infusion with 20 μg/ml hemin and 10 μg/ml folic acid) supplemented with 10% fetal bovine serum at 25 °C. Stationary-phase promastigotes were obtained from 5- to 6-day-old cultures and used throughout.

### *L. amazonensis* promastigote cultivation on 3D COL I matrices

Rat tail extracted COL I solution was diluted with 5-times concentrated DMEM media and completed to the desired density (1.5 or 3.0 mg/ml) with DMEM. 0.1 M NaOH was used to neutralize the solution pH. $10^7$ promastigotes were mixed with 1 ml of diluted and neutralized COL I solution and incubated at 37 °C for a 1 h polymerization. After complete polymerization, serum-free RPMI medium was added to the top of the matrices to feed the cells and hydrate the COL I matrix. The culture was then left at 27 °C for no more than 72 h. Viability was measured using a fluorescent Live/Dead assay (Invitrogen, L3224) and only samples with viability scores higher than 90% were used.

### Scanning electron microscopy

COL I matrix samples cultivated with *L. amazonensis* promastigotes were fixed with 2.5% glutaraldehyde in 0.1 M cacodylate buffer (pH 7.2) for 1 h and then washed overnight with PBS. After fixation, promastigotes in the COL I matrix were postfixed for 30 min in a solution containing 1% $OsO_4$, 1.25% potassium ferrocyanide and 5 mM $CaCl_2$ in 0.1 cacodylate buffer, washed in the same buffer and then dehydrated in an ethanol series from 30% to 100%. Finally, samples were critical point dried, coated with a thin gold layer in a Balzers gold sputtering system, and observed in a FEI-Quanta scanning electron microscope.

### Protease inhibitors

Metalloprotease inhibitors included 200 nM Marimastat or 5 mM o-phenantroline, cysteine protease inhibitors included 20 ng/ml Cystatin or 100 μM of trans-epoxysuccinyl-L-leucylamido-(4-Guanidino)butane (E-64) and the serine protease inhibitor used was 1 mM 4-(2-aminoethyl)-benzenesulfonylfluoride (AEBSF), (all protease inhibitors were purchased from Sigma-Aldrich). Protease inhibitors (PI) mix was used in some experiments. The PI mix used was composed of AEBSF, E-64, Cystatin and Marimastat.

## Migration assay

The *L. amazonensis* promastigote migration was characterized for all experimental conditions. Samples were placed in a culture chamber, with controlled temperature (27 °C), adapted to an inverted microscope Nikon Eclipse TE 300 (Nikon, Melville, NY). Brightfield images were captured for 5 min with a Hamamatsu C2400 CCD camera (Hamamatsu, Japan) and digitalized by a SCION FG7 frame grabber (Scion Corporation, Torrance, CA) using a frame rate of 2 frames/s. The process was repeated every 24 h for a total period of 72 h. For each movie 15 different cells were marked with black dots. The distance covered by the black dot trajectories were then obtained by image analysis using the ImageJ software (National Institutes of Health, USA) and the indirect promastigote migration rates were determined by dividing the distance covered (μm) by the time in seconds (sec). Three different movies were made and analyzed for each condition.

## Transmatrix migration (invasion) assay

For the transmatrix migration assay, the COL I matrix was previously prepared without *L. amazonensis* in a 96 well plate. Then, 100 μl serum free RPMI media containing $10^6$ promastigotes was added to the top of a 150 μl polymerized matrix. After 48 h, the RPMI media was washed out and the samples were fixed and processed for histology. The quantity of promastigotes inside the matrix (complete transmatrix process) and the distance migrated from the top to the bottom of the COL I matrix (transmatrix invasion distance) was used to determine the invasion ability of *L. amazonensis*. The transmatrix invasion distance was normalized considering 100% as the total distance between the top and the bottom of each COL I matrix histological cut. Similar results were obtained when the data were analyzed without normalization.

## COL I matrix degradation assay

*L. amazonensis* promastigotes were cultivated inside of 3D COL I matrices prepared with a solution containing 5% FITC-labeled type I Collagen (Invitrogen, Molecular Probes). The degradation assay was as modified method from *Sugiyama et al. (1980)*. After 0 h, 24 h, 48 h and 72 h, solid-phase collagens were pelleted (5 min, 16,000 g) and the supernatant containing released FITC-labeled type I Collagen fragments was measured with a spectrofluorimeter (485 nm excitation, 515 nm emission). The background signal and the total degradation were obtained from a cell-free matrix and from a matrix made with the presence of 25 U/ml type II collagenase (Gibco) respectively.

## Zymography

Promastigotes cultivated for 48 h inside COL I matrix were placed in an eppendorf tube and centrifuged for 7 min at 1600 g to separate the COL I matrix and cells from the supernatant. After centrifugation, supernatant was filtered to remove any remaining cells. Proteins from the supernatant were then separated by SDS–PAGE gel electrophoresis using gels containing 1.5 mg/ml COL I and 10% acrylamide and bis-acrylamide. After electrophoresis, gels were washed with 50 mM Tris 2.5% Triton-X100 buffer (pH 6.8) for 30 min and then incubated with 50 mM Tris, 10 mM $CaCl_2$, 1 mM DTT buffer (pH 6.8) for

48 h. When indicated, the protease inhibitors o-phenantroline 5 mM or E-64 20 µM were added to the reaction buffer.

## 3D COL matrix model for promastigote–macrophage interaction

The murine macrophages RAW 264.7 cell line ($3 \times 10^5$) were mixed with 1 ml of diluted and neutralized COL I solution and incubated at 37 °C for a 1 h polymerization. After complete polymerization, RPMI medium complemented with 5% bovine serum was added to the top of the matrices to feed the cells and hydrate the COL I matrix. The culture was then left at 37 °C for 24 h. Macrophage 3D culture viability was analyzed using a fluorescent live dye fluorescein diacetate (Sigma). After 24 h of incubation, the medium was removed and *Leishmania* promastigotes were added to the 3D culture in two ways: $10^7$ promastigotes in 200 µl of RPMI were injected inside the matrix containing macrophages (injection mode) or promastigotes were added to the top of the matrix (invasion mode).

## Confocal microscopy

Live promastigotes were labeled with 5 mM of cell tracker green (Invitrogen-C7025) for 15 min. Macrophages cultured in a 3D COL I matrix for 24 h were than labeled and put to interact with labeled *L. amazonensis* promastigotes for 4 h. After interaction the samples were fixed with 4% paraformaldehyde and than permeabilized with TritonX100 0.05% and incubated with rhodamine phalloidin (Invitrogen) overnight at 4 °C. The samples were visualized by the confocal microscopy LEICA TCS SP5 and the images were processed using the software LAS AF.

# RESULTS

## *L. amazonensis* promastigotes extensively interact with and remodel 3D COL I matrix fibers

Scanning electron microscopy (SEM) of promastigotes cultivated in a 3D COL I matrix showed a huge number of promastigotes extensively adhered to COL I fibers (Fig. 1). The presence of promastigotes altered the organization of the COL I fibers network (Figs. 1B–1F). This matrix remodeling transformed the original homogeneous COL I matrix (Fig. 1A) into a meshwork divided into areas of high fiber density and large fiber-free channels (Figs. 1E and 1F). Promastigote cell bodies and/or flagella were observed in contact with the COL I fibers (Fig. 1C). SEM revealed promastigotes completely trapped in the high density COL I fiber areas (Fig. 1D) and promastigotes in the fiber-free channels areas. Promastigotes had their cell body elongated over time during the 3D cultures on COL I matrices (Supplemental Information 1), a feature normally associated with standard liquid cultures.

## 3D COL I matrix degradation by *L. amazonensis* promastigotes

Promastigotes were cultivated inside 3D FITC-COL I matrices to measure their ability to degrade this network. *L. amazonensis* promastigotes degraded COL I fibers in a time-dependent manner and density-independent way (Figs. 2A and 2B). After 72 h, around 23% of the COL I total matrix was degraded. This degradation was significantly

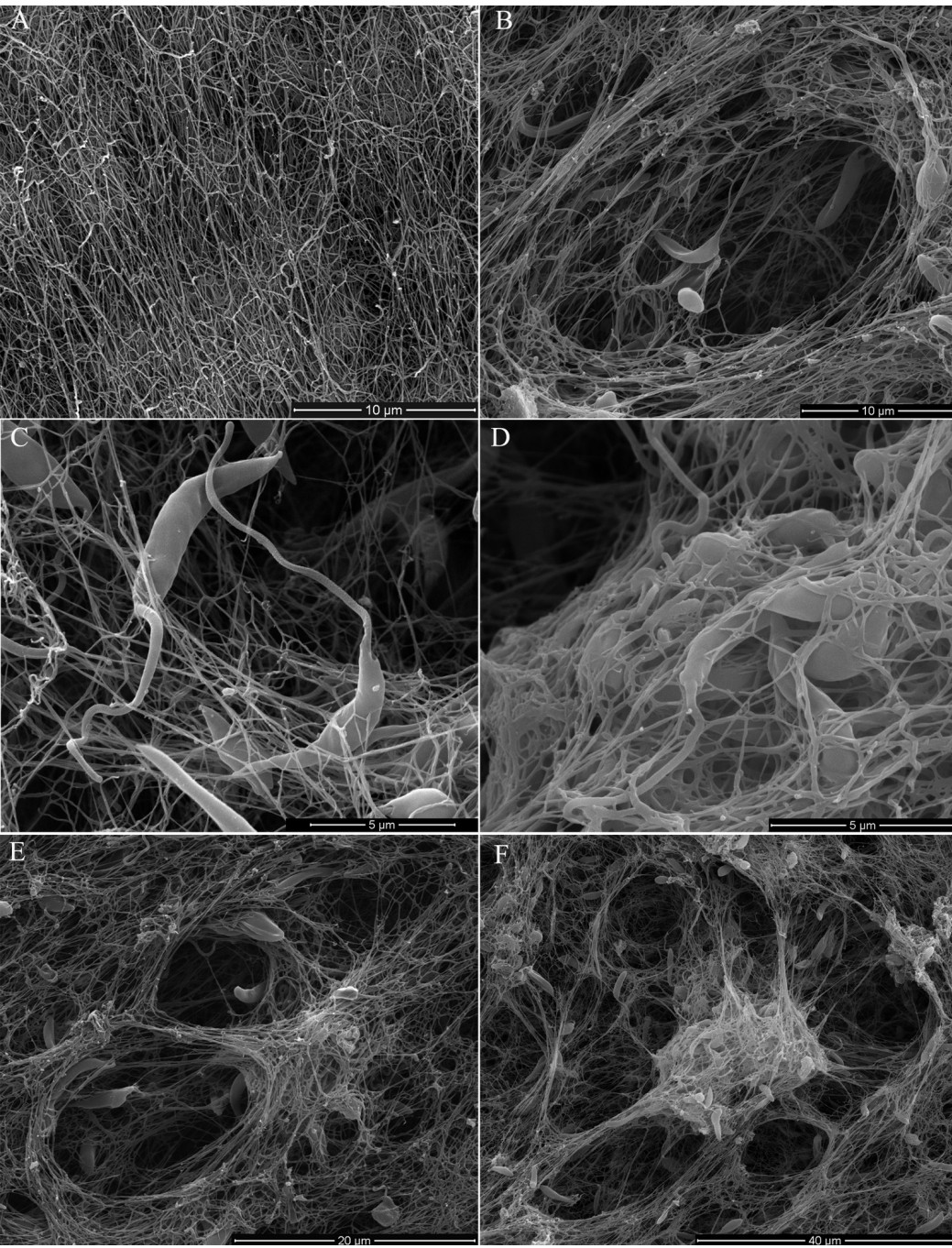

**Figure 1  SEM analysis of *L. amazonensis* interaction with COL I matrices.** SEM of normal COL I fiber organization in the 3D matrix without promastigotes (A) and promastigotes interacting with COL I fibers of a 3D matrix (B–F). Promastigotes extensive alter COL I fiber organization (B–D). In some matrix areas the *L. amazonensis* promastigotes were completely trapped between COL matrix fibers (E). *L. amazonensis* promastigotes were able to interact with COL I fibers through their cell bodies and flagella (C).

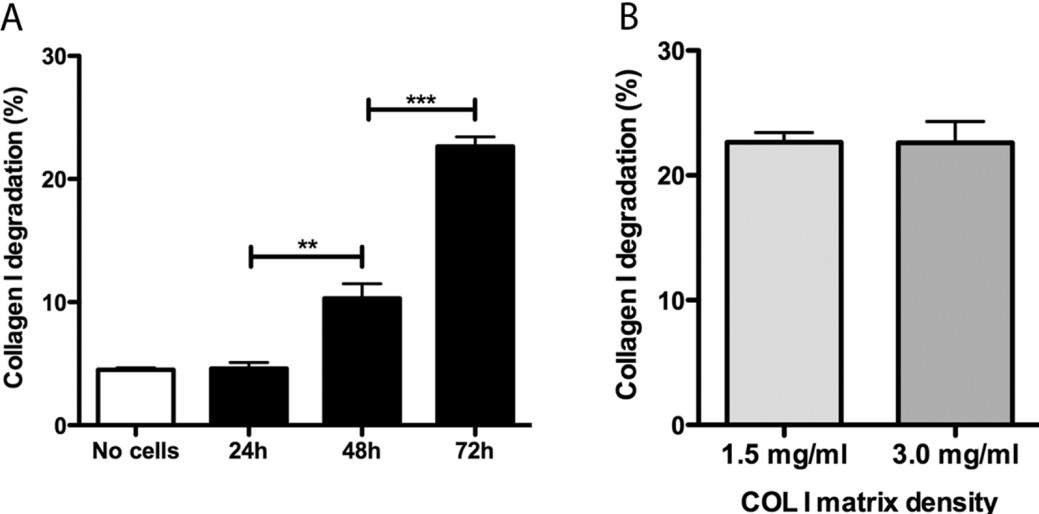

**Figure 2 Collagen matrix degradation by *L. amazonensis* promastigotes.** COL I matrix degradation was evaluated by the amount of COL I-FITC found in the supernatant after different periods of interaction of *L. amazonensis* promastigotes with COL I matrix (1.5 mg/ml) (A) or after 72 h of interaction on low (1.5 mg/ml) or high (3.0 mg/ml) COL I matrix density (B). Data are expressed as the mean with whiskers representing the standard error of the mean. *T* test with Bonferoni's multiple test correction ***, $p < 0.0001$; **, $p < 0.001$.

reduced in the presence of the cysteine protease inhibitors cystatin or E-64 (Fig. 3). The presence of the metalloprotease inhibitor marimastat also reduced COL I degradation, but no effect was observed with the serine protease inhibitor AEBSF (Fig. 3). Incubation with anti-gp63 caused a small but significant reduction of COL I degradation, suggesting that this metalloprotease is also capable of COL I cleavage. COL degradation was not completely abolished in the presence of protease inhibitor (PI) mix (Fig. 3). Although the concentration of inhibitors used could be insufficient for complete inhibition, we chose not to use higher concentrations of this inhibitor mix because they affected cell viability (Table S1). Zymography of culture supernatant from *L. amazonensis* grown in a COL I matrix detected two collagenase protease bands (Supplemental Information 2). The collagenase bands completely disappeared in the presence of the metalloprotease inhibitor O-phenanthroline, demonstrating the secretion of COL I-degrading metalloproteases by *L. amazonensis* promastigotes (Supplemental Information 2). Other classes of proteases could not be detected by COL zymography in the conditions tested. However, reaction buffers of pH lower than 6.8 destroy the COL gels, restricting the use of this assay for *Leishmania* cysteine proteases.

### *L. amazonensis* migration and transmatrix migration in a 3D COL I matrix

*L. amazonensis* promastigote migration inside of a 3D COL I matrix was observed by video microscopy (Video S1). These flagellated protozoa displayed an intermittent migration mode with displacement periods (run phase) intercalated with non-displacement periods

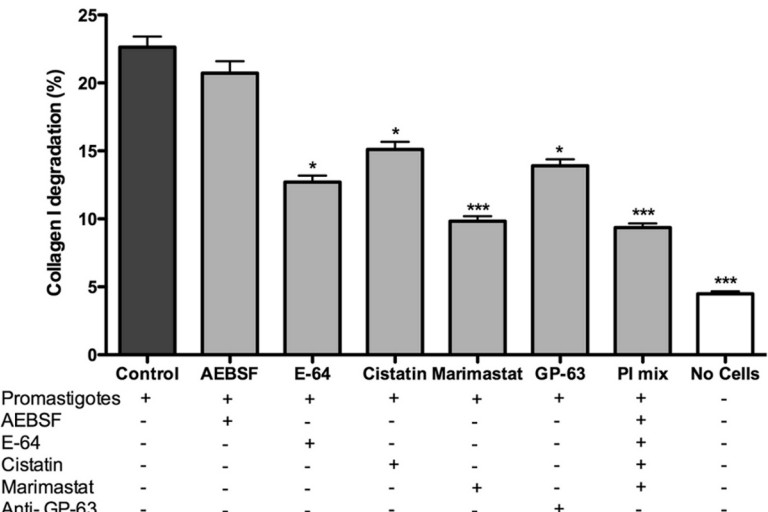

**Figure 3** *L. amazonensis* **promastigotes degrade COL I fiber by protease release.** Promastigote inter-action with the COL I matrix occurred in the presence of different proteases inhibitors (PI) and PI mix. Cysteine PI, E-64 (100 μM) and cystatin (20 ng/ml), caused a significant reduction of COL I degradation, however, the metalloprotease inhibitor marimastat (200 nM) caused an even higher reduction. The presence of anti GP-63 (1:50) also reduced COL I degradation ability. Serine protease inhibitor AEBSF (1 mM) had no effect on the degradation. Whiskers represent the standard error of the mean. 1-way ANOVA, Kruskal–Wallis test, *, $p < 0.05$; ***, $p < 0.0001$.

(freeze phase). The freeze and run migration mode contrasts with the continuous swimming migration mode found in conventional *in vitro* liquid culture. The migration speed in 3D cultures significantly increased over time (Fig. 4), reaching the highest mean rate (6 μm/s) after 72 h. Even at the first time point, just after COL I matrix polymerization, promastigotes had a slow but non-zero migration rate (Fig. 4). Over time, the displacement periods were more frequent and lasted longer, reflecting a higher migration speed (Fig. 4). Thus, there was a correlation between time, increased speed, and COL I matrix remodeling by promastigotes.

In addition to migration within a 3D COL I matrix, we tested the ability of pro-mastigotes to adhere to and cross the matrix (transmatrix migration). Indeed, when promastigotes were added to the top of a polymerized 3D COL I matrix they were able to adhere to and penetrate through the COL I fiber meshwork (Fig. 5A). This matrix invasion ability (transmatrix migration) demonstrates that promastigotes are not only able to migrate inside of a 3D COL matrix (Fig. 4) but are also able to cross from a liquid environment (RPMI media) to a 3D meshwork (COL I matrix). Here it is important to note that promastigote transmatrix migration was observed without the addition of an external chemo-attractive factor, demonstrating a natural COL I matrix invasive ability for this intracellular parasite. The distance reached by promastigotes from the top of the matrix (transmatrix migration distance) was significantly reduced at higher COL I densities (3.0 mg/ml) compared to lower densities (1.5 mg/ml) (Fig. 5B).

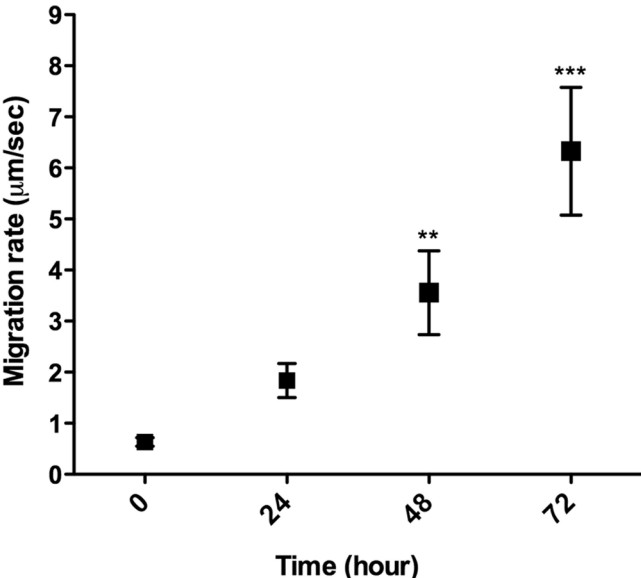

**Figure 4 3D migration rate increases over the time in COL I matrix.** *L. amazonensis* promastigote migration rate inside of COL I 3D matrix over time. The migration was observed by videomicroscopy and the rate analyzed by ImageJ software. Bars represent the standard error of the mean. 1-way ANOVA, **, $p < 0.001$; ***, $p < 0.0001$.

## Extracellular protease activity promotes *L. amazonensis* 3D COL I migration and invasion

The correlation between protease activity and promastigote migration in 3D was analyzed by videomicroscopy in the presence or absence of a protease inhibitor (PI) cocktail. The promastigote migration rates were analyzed using the software ImageJ. In the presence of a PI mix composed of E-64, Marimastat and Cystatin, the promastigote migration rates inside the 3D COL I matrix were significantly reduced, demonstrating that protease activity increases migration rates (Fig. 6). The fraction of promastigotes with migration rates exceeding 5 μm/s was also significantly decreased by the PI mix (Fisher's exact test, Fig. 6). The reduction of migration rates could also be observed when only the run phase was considered in the analysis, indicating an inhibition of both frequency and maximum speed of migration.

In addition, the role of protease activities in promastigote transmatrix migration was evaluated after 72 h, in the presence or absence of PI mix (Fig. 7). Transmatrix migration is dependent on two processes: the ability of cells to penetrate into the matrix and, once inside, the ability to migrate. The PI mix presence reduced by almost 40% the number of promastigotes inside of the COL I matrix (Fig. 7A) and also significantly reduced the maximal transmatrix migration distance (in depth) achieved by the promastigotes (Fig. 7B).

Together the data demonstrate that extracellular protease activity is important for *L. amazonensis* promastigote transmatrix and migration in 3D cultures (Figs. 6 and 7), due to COL I matrix degradation/remodeling.

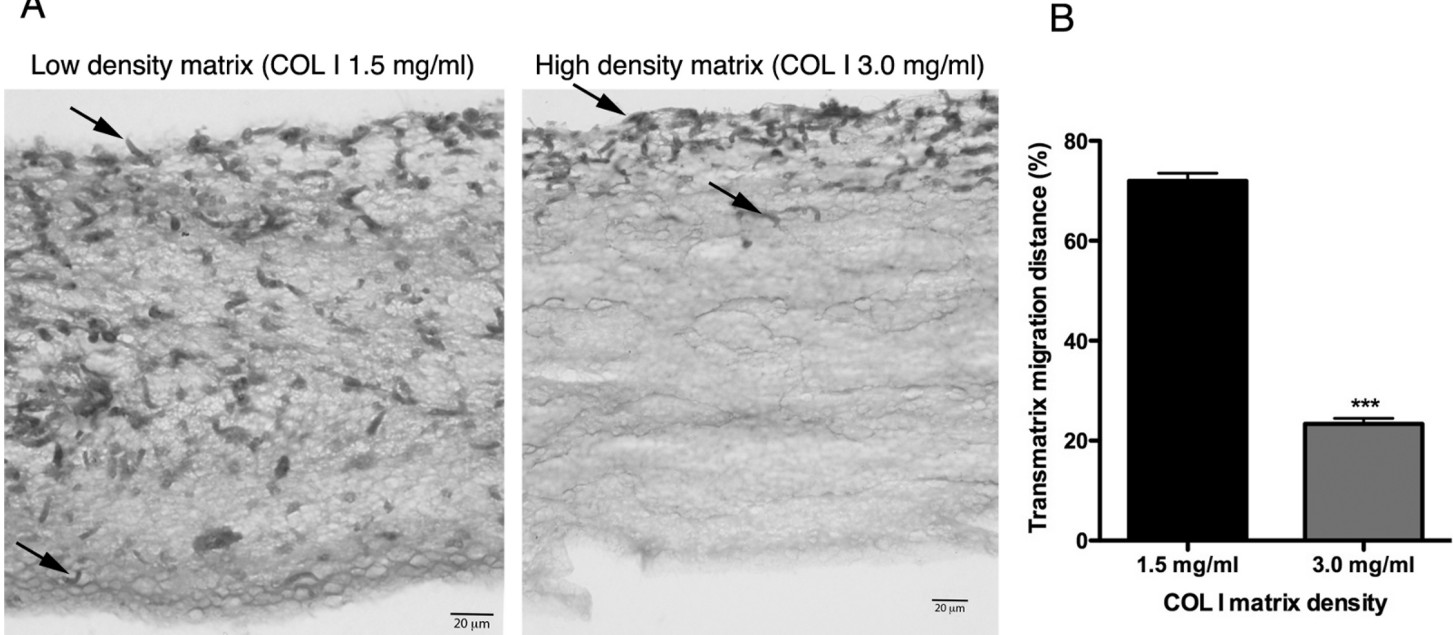

**Figure 5 Trans-matrix migration ability is dependent on the COL I matrix density.** Histological transversal cuts of COL I 3D matrix at 1.5 or 3.0 mg/ml after 72 h of *L. amazonensis* promastigote (arrows) invasion (A). Promastigote COL I transmatrix migration was measured by the depth reached by the promastigotes divided by the total size of each histological transversal cut (B). Whiskers represent the standard error of the mean. Student's *T* test ***, $p < 0.0001$.

## Interaction of macrophages and promastigotes in 3D COL I matrix

For an even better mimic of what would happen during *Leishmania* infection, we cultivated macrophages inside the COL I matrix for 24 h, after which *L. amazonensis* promastigotes were added to the top of the matrix (Fig. 8). After 6 h of interaction most of the parasites were found inside the matrix (Fig. 8), a shorter time than in the macrophage-free matrix. This suggests an influence of macrophages on the invasion ability of *Leishmania* promastigotes. 3D visualization by confocal microscopy of macrophage-promastigote interaction (Figs. 9A and 9B) showed macrophages possessing a round morphology with many actin-membrane filament projections (Fig. 9C). In some cases these thin actin-membrane projections were touching a promastigote's surface (Fig. 9D). SEM microscopy observations showed the extensive contact of macrophages and promastigotes with COL I fibers in the 3D culture model (Fig. 10). The fibrotic characteristic of the ECM 3D environment seems to create a effective barrier for physical contact between *Leishmania* and macrophages, in contrast to 2D culture models in which macrophages become infected much earlier.

The injection of promastigotes directly into the matrix containing macrophages allowed a good distribution of promastigotes inside the COL meshwork and enabled the observation of macrophage-promastigote interaction in the first minutes of interaction (Video S2). That experiment demonstrated the promastigotes migrating further and faster

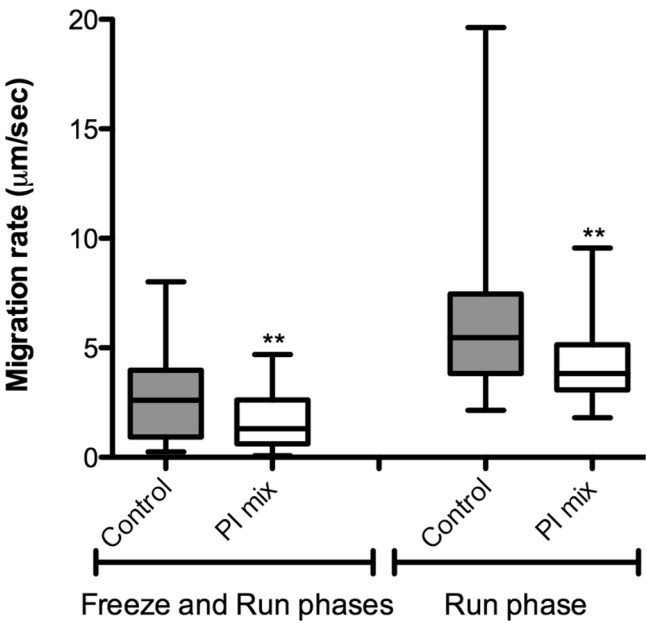

**Figure 6** *L. amazonensis* **migration in 3D cultures with and without protease inhibitors.** *L. amazonensis* promastigote migration rate after 24 h inside of COL I 3D matrix with or without the protease inhibitors (PI) mix. The migration was observed by videomicroscopy and analyzed by ImageJ software. The migration rate was analyzed in two ways: total migration (freeze and run phases) or during the moving phase of the migration only (run phase). Whiskers represent the minimum and maximum values. Differences in the mean value were tested using Mann–Whitney test **, $p = 0.0074$ and $p = 0.0019$; $n = 45$. Fisher's exact test was used to test differences in the fraction of promastigotes with migration speed greater than 5 μm/s, $p = 0.0124$ (Freeze and run phases) and $p = 0.0031$ (run phase).

than macrophages in the 3D culture. Additionally, the promastigotes exhibited the same freeze and run migration mode observed in the macrophage-free 3D matrix.

## DISCUSSION

In the human host, *Leishmania amazonensis* is an intracellular parasite that mainly proliferates inside macrophages. Therefore, the interaction of the *Leishmania* parasite with the host ECM has been neglected. In fact, it is critical to understand the interaction of *Leishmania* promastigotes with the ECM because when a host is infected, the promastigotes must pass through the dermis ECM, remaining there until the first contact with macrophages (*Lira, Rosales-Encina & Arguello, 1997*; *McGwire, Chang & Engman, 2003*) or other potential host cells. Our results demonstrated that *L. amazonensis* promastigotes extensively contacted the COL I fibers in a 3D matrix (Fig. 1D). Following these interactions were drastic modifications of the COL I fiber organization (Figs. 1A–1F). Many publications have shown a correlation between ECM remodeling and inflammation, and indeed, abnormal ECM organization is prominent in many diseases such as tissue fibrosis and cancer (*Cox & Erler, 2011*). This suggests that COL I matrix remodeling by promastigotes could play an important role during the infection process (*Larsen et al., 2006*; *Stamenkovic, 2003*).

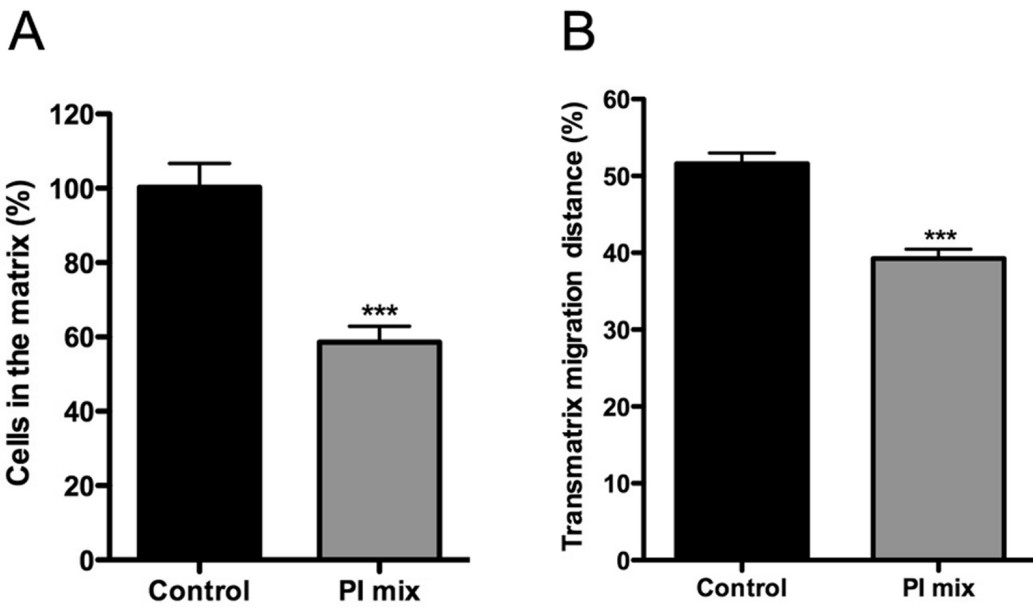

**Figure 7 Trans-matrix migration ability is affected by the presence of a protease inhibitors (PI) mix.** *Leishmania* promastigotes were added to the top of COL I matrix of 1.5 mg/ml and the trans-matrix invasion ability in presence or absence of protease inhibitors was analyzed by the percentage of parasites inside the matrix (A) and by the maximum distance (in depth) achieved by the parasites (B). In both cases the presence of protease inhibitors mix significantly reduced the transmatrix migration ability.

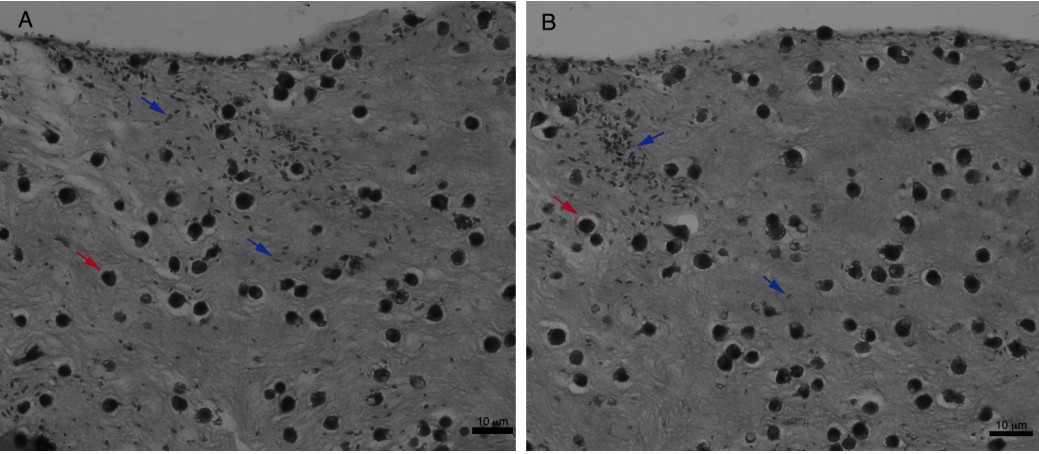

**Figure 8 Promastigote trans-matrix migration ability in the presence of macrophages.** *Leishmania* promastigotes were added to the top of the macrophages 3D COL I culture. After 6 h the samples were fixed, prepared for histology and transversally cut. Image shows transversal cuts of COL I matrix containing macrophages (red arrows) 6 h after addition of *L. amazonensis* (blue arrows) promastigotes (A–B) demonstrating many parasites able to invade the COL I matrix.

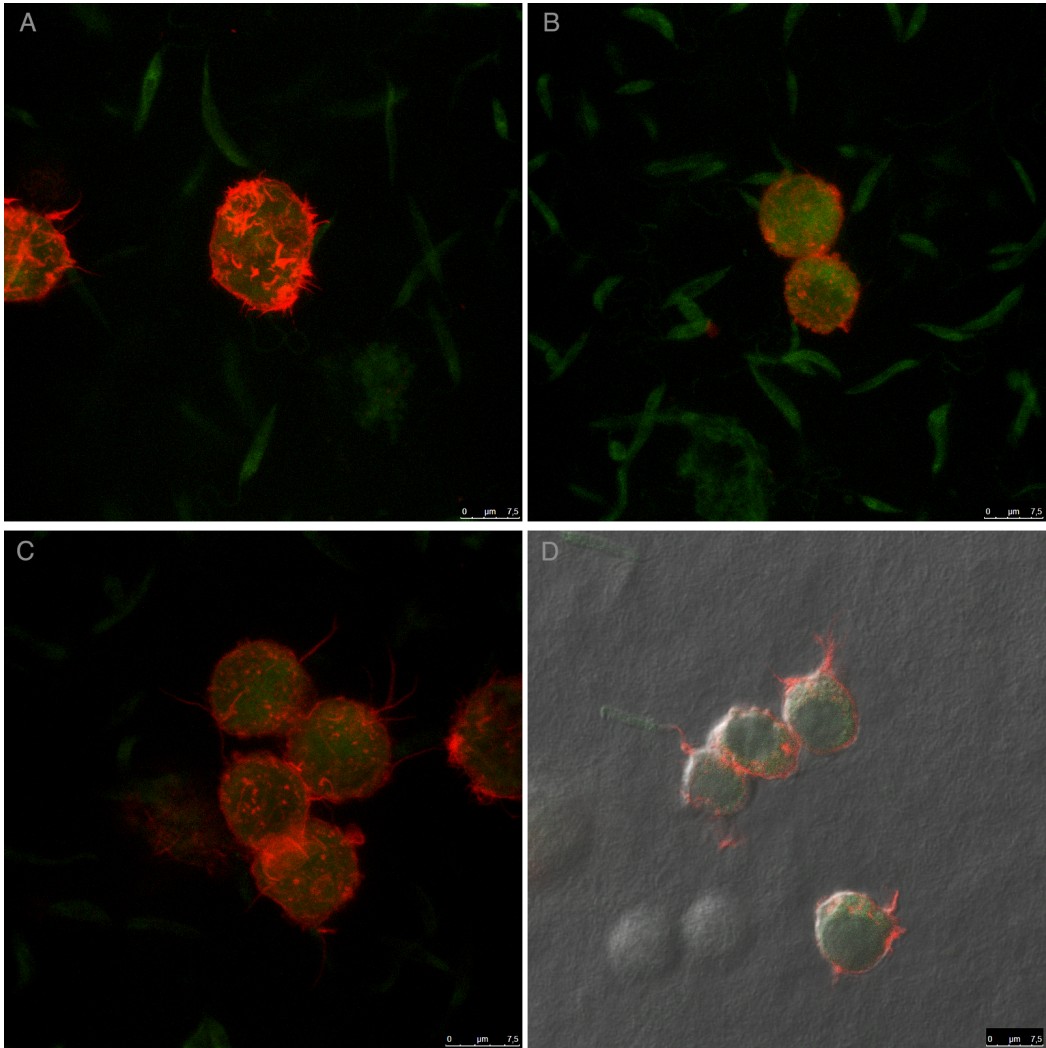

**Figure 9 Macrophage–*Leishmania* 3D interaction vizualization by confocal microscopy.** *Leishmania* interaction with macrophages in the 3D COLI matrix was visualized by confocal microscopy. Live *Leishmania* promastigotes were labeled with cell tracker green before interaction. Macrophage actin filaments were stained with phalloidin (red). Images are 3D reconstruction of a 11 µm deep image.

3D culture is becoming a common approach for the study of several types of mammalian cells (*Voytik-Harbin, 2001*) but it is still not commonly used for pathogens or unicellular organisms (*Behnsen et al., 2007*). In this work we establish, for the first time, 3D *in vitro* cultivation of *L. amazonensis* promastigotes. This *in vitro* 3D culture mimics (at least in part) the physical dynamics of the *Leishmania* human host environment and makes possible the *in vitro* study of the mechanisms involved in the interaction of *L. amazonensis* with the ECM. Conventionally, promastigotes are cultivated in rich serum-containing media to maintain 90% viability. However, in our model, *L. amazonensis* promastigotes were able to survive ($>$90% viability) and proliferate for 72 h inside a 3D matrix composed of 1.5 or 3.0 mg/ml of COL I in serum-free RPMI medium, suggesting that this flagellated parasite, like some mammalian cells, required fewer survival signals

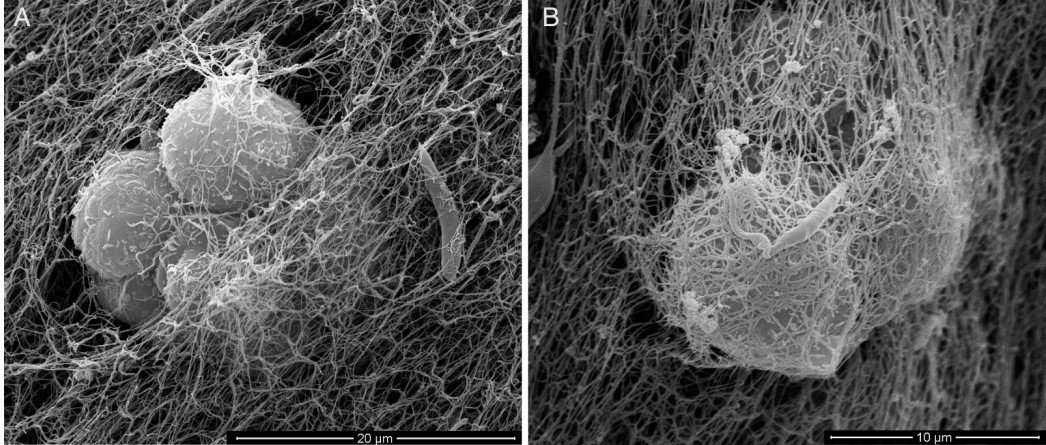

**Figure 10 SEM of *L. amazonensis*–macrophages 3D interaction.** SEM visualization of macrophages cultured in a 3D COL I matrix after 3 h (A) or 4 h (B) of interaction with *L. amazonensis* promastigotes. In order to visualize the promastigote–macrophage interaction inside the matrix the samples were transversally cut in liquid nitrogen during SEM samples processing.

when in 3D culture (*Alavi & Stupack, 2007*; *Eke & Cordes, 2011*). This could be due to the physical support presented in the 3D matrix or to the chemical signaling provided by the COL I molecules (*Weaver et al., 2002*).

The roles that *Leishmania* proteases play during inflammation and interaction with macrophages are important for infection, especially in the case of the metalloprotease gp63 (*Silva-Almeida et al., 2012b*; *Yao, 2010*). Here, we presented a new role for extracellular *Leishmania* proteases. Protease blockage by a mixture of inhibitors decreased matrix degradation and affected promastigote migration and transmigration among COL I fibers. The ability of promastigotes to cleave COL I matrix fibers was analyzed by a degradation assay using fluorescent COL I. We showed that promastigotes can actively degrade COL I fibers and that the degradation was independent of the matrix density (Fig. 2). Metallo- and cysteine proteases were implicated in this degradation (Fig. 3). The presence of anti-gp63 significantly reduced the COL I degradation, indicating the importance of this protease in the cleavage of COL I fibers. However, the presence of the PI mix did not completely abolish the COL I degradation. These findings suggest that either the inhibitors are not strong enough to block all the extracellular proteases without affecting the parasite survival, or that the mechanical pushing and pulling by the promastigote flagella also contribute to COL I fiber degradation.

Following the promastigotes by videomicroscopy, we showed that these parasites were able to migrate inside the 3D COL I matrix (Fig. 4). However, the migration mode in this 3D culture was completely different from the one in standard culture medium. In the 3D culture environment, the promastigote migration oscillates between periods of movement and stationary periods, which we described as a freeze and run migration mode (Video S1). This data is consistent with our SEM data (Fig. 1), showing some promastigotes completely trapped in the COL I fibers (freeze phase) and others COL I fiber-free

(run phase), probably passing through the COL I matrix tunnels created by matrix remodeling (Fig. 1). The migration rate and COL I matrix degradation both increased over time (Figs. 2–4), consistent with the hypothesis that the run phase of migration is linked to the presence of COL I matrix tunnels.

In the presence of PI mix the rate of promastigote migration inside the matrix was significantly reduced (Fig. 6). This reduction was more pronounced when only the run phase was considered; the maximum speed reached in that phase was significantly slower in the presence of PIs (Fig. 6). These results suggest that the extracellular *Leishmania* proteases and the subsequent COL I matrix remodeling lead directly to an increased migration rate.

Transmatrix migration is an important skill of metastatic cells, invasive extracellular parasites, and of immune cells such as neutrophils and macrophages (*Hagedorn & Sherwood, 2011*; *Schoumacher, Louvard & Vignjevic, 2011*). *L. amazonensis* promastigotes were able to invade matrices of multiple COL I densities (Fig. 5), showing that intracellular parasites possess the ability not only to migrate inside of a COL I matrix but also to adhere to and invade these matrices (transmatrix migration), raising the unexpected possibility of invasive behavior by this intracellular parasite during infection. It is important to mention that all our experiments were made with stationary phase promastigotes (Methods) not enriched for metacyclics and we do not have access to the metacyclic/procyclic proportion of our promastigote culture. Therefore, it is possible that these forms have different behavior in 3D cultures and further work is needed to evaluate their roles.

Taken together, we showed that promastigotes degrade COL I matrix via extracellular cysteine and metalloprotease, and these proteases play a role during migration and invasion of 3D matrices. Together these results suggest that the matrix remodeling contributes to *L. amazonensis* migration and invasion by helping promastigotes to get free from the COL I matrix traps (reducing the freeze period), or even to open the tunnels inside of the matrix that enable a faster maximum migration speed. In both ways, protease release and the subsequent matrix remodeling could help the *L. amazonensis* promatigotes to create a 3D environment that facilitates access to their intracellular "home".

Furthermore, we establish a 3D co-culture model with macrophages seeded in a COL I matrix for 24 h before the addition of *L. amazonensis* promastigotes. The promastigotes were added in two ways: at the top of the matrix (invasion model) or by syringe injection directly inside the matrix. Both models resulted in direct *Leishmania*–macrophage contact inside the 3D environment. However, the invasion model was necessary for the analysis of the trans-matrix migration of promastigotes during interaction (Fig. 8), while the injection model created a more homogenous distribution of promastigotes in the matrix, allowing videomicroscopy of *Leishmania*–macrophages interactions (Video S2). Surprisingly, promastigotes migrated faster than macrophages in this model, supporting an active participation of promastigotes during *Leishmania* infection. The *Leishmania*–macrophage 3D interaction assay established here will help us to better understand some aspects of *Leishmania*–macrophage and *Leishmania*-matrix interaction and could become an important model for initial drug screens.

**Abbreviations**

| | |
|---|---|
| **COL I** | Collagen I |
| **3D** | three-dimensional |
| **ECM** | extracellular matrix |
| **FITC** | fluorescein isothiocyanate |
| **PI** | protease inhibitors |
| **SEM** | scanning electron microscopy |

## ACKNOWLEDGEMENTS

We thank Dr. Sherri Smith and Dr. Robin C. Friedman for helpful comments on the manuscript.

### Funding

This work was supported by Conselho Nacional de Desenvolvimento Científico e Tecnologico (CNPq), Fundação Carlos Chagas Filho de Amparo à Pesquisa do Estado do Rio de Janeiro (FAPERJ), Coordenação de Aperfeiçoamento de Pessoal de Nível Superior (CAPES), Programa de Apoio à Núcleos de Excelência (PRONEX), "Apoio ao Estudo de Doenças Negligenciadas e Emergentes" from the FAPERJ and Instituto Nacional de Ciência e Tecnologia de Fluidos Complexos (INCT-FCx). The funders had no role in study design, data collection and analysis, decision to publish, or preparation of the manuscript.

### Grant Disclosures

The following grant information was disclosed by the authors:
Conselho Nacional de Desenvolvimento Científico e Tecnologico (CNPq).
Fundação Carlos Chagas Filho de Amparo à Pesquisa do Estado do Rio de Janeiro (FAPERJ).
Coordenação de Aperfeiçoamento de Pessoal de Nível Superior (CAPES).
Programa de Apoio à Núcleos de Excelência (PRONEX).
"Apoio ao Estudo de Doenças Negligenciadas e Emergentes" from the FAPERJ.
Instituto Nacional de Ciência e Tecnologia de Fluidos Complexos (INCT-FCx).

### Competing Interests

The authors declare there are no competing interests.

### Author Contributions

- Debora B. Petropolis conceived and designed the experiments, performed the experiments, analyzed the data, wrote the paper, prepared figures and/or tables, reviewed drafts of the paper.
- Juliany C.F. Rodrigues conceived and designed the experiments, contributed reagents/materials/analysis tools, wrote the paper, reviewed drafts of the paper.

- Nathan B. Viana conceived and designed the experiments, analyzed the data, contributed reagents/materials/analysis tools, reviewed drafts of the paper.
- Bruno Pontes performed the experiments, analyzed the data, wrote the paper, reviewed drafts of the paper.
- Camila F.A. Pereira performed the experiments.
- Fernando C. Silva-Filho contributed reagents/materials/analysis tools, wrote the paper.

## Ethics

The following information was supplied relating to ethical approvals (i.e., approving body and any reference numbers):

The use of animal models was approved by the Ethics Committee for Animal Experimentation of the Health Sciences Centre, Federal University of Rio de Janeiro (Protocols n. IBCCF 096/097), according to the Brazilian federal law. All animals received humane care in compliance with the "Principles of Laboratory Animal Care" formulated by the National Society for Medical Research and the "Guide for the Care and Use of Laboratory Animals" prepared by the National Academy of Sciences, USA.

## Supplemental Information

Supplemental information for this article can be found online at http://dx.doi.org/10.7717/peerj.317.

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

# PeerJ

**Hagedorn EJ, Sherwood DR. 2011.** Cell invasion through basement membrane: the anchor cell breaches the barrier. *Current Opinion in Cell Biology* **23**:589–596 DOI 10.1016/j.ceb.2011.05.002.

**Hynes RO. 2009.** The extracellular matrix: not just pretty fibrils. *Science* **326**:1216–1219 DOI 10.1126/science.1176009.

**Kaye P, Scott P. 2011.** Leishmaniasis: complexity at the host–pathogen interface. *Nature Reviews Microbiology* **9**:604–615 DOI 10.1038/nrmicro2608.

**Kulkarni MM, Jones EA, McMaster WR, McGwire BS. 2008.** Fibronectin binding and proteolytic degradation by Leishmania and effects on macrophage activation. *Infection and Immunity* **76**:1738–1747 DOI 10.1128/IAI.01274-07.

**Larsen M, Artym VV, Green JA, Yamada KM. 2006.** The matrix reorganized: extracellular matrix remodeling and integrin signaling. *Current Opinion in Cell Biology* **18**:463–471 DOI 10.1016/j.ceb.2006.08.009.

**Lira R, Rosales-Encina JL, Arguello C. 1997.** *Leishmania mexicana*: binding of promastigotes to type I collagen. *Experimental Parasitology* **85**:149–157 DOI 10.1006/expr.1996.4127.

**McGwire BS, Chang KP, Engman DM. 2003.** Migration through the extracellular matrix by the parasitic protozoan Leishmania is enhanced by surface metalloprotease gp63. *Infection and Immunity* **71**:1008–1010 DOI 10.1128/IAI.71.2.1008-1010.2003.

**McMaster WR, Morrison CJ, MacDonald MH, Joshi PB. 1994.** Mutational and functional analysis of the Leishmania surface metalloproteinase GP63: similarities to matrix metalloproteinases. *Parasitology* **108**(**Suppl**):S29–S36 DOI 10.1017/S0031182000075697.

**Melo FA, Moura EP, Ribeiro RR, Alves CF, Caliari MV, Tafuri WL, Calabrese KS, Tafuri WL. 2009.** Hepatic extracellular matrix alterations in dogs naturally infected with Leishmania (Leishmania) chagasi. *International Journal of Experimental Pathology* **90**:538–548 DOI 10.1111/j.1365-2613.2009.00681.x.

**Murray HW, Berman JD, Davies CR, Saravia NG. 2005.** Advances in leishmaniasis. *The Lancet* **366**:1561–1577 DOI 10.1016/S0140-6736(05)67629-5.

**Neuber H. 2008.** Leishmaniasis. *Journal der Deutschen Dermatologischen Gesellschaft* **6**:754–765 DOI 10.1111/j.1610-0387.2008.06809.x.

**Rogers ME. 2012.** The role of Leishmania proteophosphoglycans in sand fly transmission and infection of the mammalian host. *Frontiers in Microbiology* **3**:223 DOI 10.3389/fmicb.2012.00223.

**Schoumacher M, Louvard D, Vignjevic D. 2011.** Cytoskeleton networks in basement membrane transmigration. *European Journal of Cell Biology* **90**:93–99 DOI 10.1016/j.ejcb.2010.05.010.

**Silva-Almeida M, Carvalho LO, Abreu-Silva AL, Souza CS, Hardoim DJ, Calabrese KS. 2012a.** Extracellular matrix alterations in experimental *Leishmania amazonensis* infection in susceptible and resistant mice. *Veterinary Research* **43**:10 DOI 10.1186/1297-9716-43-10.

**Silva-Almeida M, Pereira BA, Ribeiro-Guimaraes ML, Alves CR. 2012b.** Proteinases as virulence factors in *Leishmania* spp. infection in mammals. *Parasites & Vectors* **5**:160 DOI 10.1186/1756-3305-5-160.

**Stamenkovic I. 2003.** Extracellular matrix remodelling: the role of matrix metalloproteinases. *Journal of Pathology* **200**:448–464 DOI 10.1002/path.1400.

**Sugiyama K, Yamamoto K, Kamata O, Katsuda N. 1980.** A simple and rapid assay for collagenase activity using fluorescence-labeled substrate. *Kurume Medical Journal* **27**:63–69 DOI 10.2739/kurumemedj.27.63.

**Voytik-Harbin SL. 2001.** Three-dimensional extracellular matrix substrates for cell culture. *Methods in Cell Biology* **63**:561–581 DOI 10.1016/S0091-679X(01)63030-9.

**Weaver VM, Lelievre S, Lakins JN, Chrenek MA, Jones JC, Giancotti F, Werb Z, Bissell MJ. 2002.** beta4 integrin-dependent formation of polarized three-dimensional architecture confers resistance to apoptosis in normal and malignant mammary epithelium. *Cancer Cell* **2**:205–216 DOI 10.1016/S1535-6108(02)00125-3.

**Yao C. 2010.** Major surface protease of trypanosomatids: one size fits all? *Infection and Immunity* **78**:22–31 DOI 10.1128/IAI.00776-09.

**Yao C, Leidal KG, Brittingham A, Tarr DE, Donelson JE, Wilson ME. 2002.** Biosynthesis of the major surface protease GP63 of Leishmania chagasi. *Molecular and Biochemical Parasitology* **121**:119–128 DOI 10.1016/S0166-6851(02)00030-0.