# Peer review of "Leishmania amazonensis promastigotes in 3D Collagen I culture: an in vitro physiological environment for the study of extracellular matrix and host cell interactions"

_PeerJ, doi:10.7717/peerj.317_

## Round 0.1 · original submission · Major Revisions

The reviewers showed interests in the manuscript and made some general and specific comments. You are invited to revise your manuscript according to these comments and submit your responses to these comments at a point-to-point basis.

Reviewer 1 ·

Basic reporting

The authors present data on the establishment of L. amazonensis promastigote culture in a 3D environment mainly composed by collagen I to study L. amazonensis-extracellular matrix relationship.

The manuscript is clearly written, and requires only some editing to correct a few scattered errors.

Major criticisms:

The title is not very informative; it expresses only a part of the work. Perhaps a more accurate and descriptive title will be more appropriate.

Leishmania and Leishmania amazonensis must be in italic.

The three first paragraphs of the introduction could be deleted (Leishmaniasis… Neuber 2008. Line 18 to 25). This information is very ordinary. Introduction could get straight on the subject of the manuscript;

Insert the reference “Silva-Almeida et al. 2012a” together with Abreu-Silva et al. 2004 and Melo et al. 2009. Line 31);

Line 53 to 59 is results, not introduction;

Line 80- BALB/c instead Balb/C;

Line 175- It is not possible to see dividing promastigotes in figure 1B. please provide a high magnification or rewrite the sentence;

Line 293- Videomicroscopy used to show parasite migration within 3D COL I matrix was one of the supporting video sent (2FPS_20X_1_(CROP)2MIN.AVI.ZIP or MOVIE_24H_2FPS_C.AVI.ZIP)?

If yes, it is not possible to see the parasite migration. I am sure the data is corrected but the video is unnecessary.

Line 297- supplement 3 and Line 332- supplement 4 was not attached in the journal page.

Figure 1 Legend- Magnification of figure F is tiny. It is not possible to visualize the parasite-COL I fiber interaction.

Figure 8- Insert the letters “A” and “B” in the legend.

An English revision could be done by a native speaker.

Experimental design

The experimental design is well established but we have a question:
Figure 2- Which is the COL I matrix density used in the experiment expressed in figure A?

Validity of the findings

The manuscript is very interesting and the data from the basis are important to contribute to the Leishmania-extracellular matrix interaction knowledge.

Additional comments

The manuscript is very interesting and contribute to field of study.

Reviewer 2 ·

Basic reporting

no comments

Experimental design

All the experimental suggestions were cited along the PDF.

Validity of the findings

no comments

Additional comments

The paper is extremely interesting and well-structured. However, some relevant points need to be addressed by the authors before further considering the paper for publication. In this context, all the critics, doubts and suggestions were highlighted in yellow along the PDF.

Annotated reviews are not available for download in order to protect the identity of reviewers who chose to remain anonymous.

Reviewer 3 ·

Basic reporting

No Comments

Experimental design

No Comments

Validity of the findings

No Comments

Additional comments

Review

Leishmania amazonensis 3D invasion and migration promoted by extracellular proteases

This is a very interesting work on Leishmania cell biology, concerned about the early steps of parasite establishment in mammalian hosts, i.e., the transfer of promastigotes from invertebrate to vertebrate hosts. This is a poorly understood topic regarding Leishmania infection and papers on this matter are welcome. The manuscript also present a collagen I cell substrate, employed to mimic mammalian host dermis and to evaluate the invasion and migration of infective parasite populations in this condition. However, considering the 10-years old works of McGwire (2002, 2003), the present study should be improved to diverge or complement these previous works and to increase its relevance to the field.

Major issues:

1- Low-infective stages (procyclics) must be compared to stationary phase promastigotes in Figure 2A. Perhaps metacyclic-enriched populations should also be compared. Is collagen I (COLI) degradation a bona fide "virulence" mechanism or non-infective promastigotes are also able to degrade COLI? McGwire et al, 2002 shows that gp63, for example, is not released from avirulent L. amazonensis and suggests "an association of gp63 release with virulence". Is it the case for the proteases involved in collagen I degradation?

2- The assumption that COLI degradative proteases are released from promastigotes (extracellular proteases) needs to be further investigated. The presented results do not exclude the possibility that these proteases are mainly associated to promastigote surface. The zymography is not a straightforward method to test the influence of extracellular proteases in COLI degradation: why not directly test promastigote supernatants (SN) on COLI-FITC degradation? Promastigote culture SN (from COLI matrix or conventional cultures) collected in log-, stationary- and death-phases (or in 1, 2 or 3 days of cultivation, in the case of COLI matrix) might display different COLI degradation efficiencies.

3- There is some misconception regarding the use of 3D terminology. Figure 4 and Figure 6 are examining a 2D migration in a 3D matrix, what is different from 3D migration studies. 3D migration would be to evaluate space and distances in x, y and z coordinates of the matrix. The present study evaluates migration only and always in two dimensions, i.e., distance (x,y) in single focal planes of videomicroscopy or migration in planes of COLI matrix transversal cuts (x, z). 3D migration means an integrative analysis between x,y and z distances, not performed in the study. Thus, the manuscript should be modified in several instances, including the title (which is misleading). May I suggest something like "Promastigotes invasion and migration through Collagen I Matrices is promoted by proteases in vitro"?

There is also misunderstanding about what a "3D environment"(line 295) is. Animals, plants, insects, cells and parasites, we all live in three-dimensional environments. An in vitro culture model of macrophages and Leishmania is not a "2D culture model" (line 246) but a micrometric 3D environment with a lot of spatial information. COLI matrices are as three-dimensional as promastigotes plus macrophages systems, so there is a lot of redundancy in "3D COLI matrix". Authors should use 3D terminology more carefully and accurately.

4- Results on promastigote migration speed (not trans-matrix migration) in the presence of protease inhibitors (PI) are problematic, in the opinion of this reviewer. Figure 4 shows promastigote speed (um/sec) after 24, 48 and 72h of cultivation on COLI matrices - with best results after 72h of cultivation. There is no significant difference between time point 0h and 24h, suggesting that after 24h of cultivation COLI matrices are not disrupted enough to facilitate promastigote migration. However, in Figure 6 (where PI tests are presented) promastigote speed was evaluated after 24h of cultivation on matrices - based on Figure 4, 24h is not the adequate time point for evaluate migration on COLI matrices in the presence of PI. Is PI cytotoxicity the reason for choosing this time point? Figure 3 results were also obtained after 24h of cultivation on matrices? (Authors must include this important information in Figure 3). If cytotoxicity effects are observed with prolonged incubation of PI and promastigotes, why results of Figure 7 were obtained after 72h (line 227) in the presence of PI mix and results of Figure 4 after 24h?

Additionally, based on Figure 6 box plot, it is suggestive that the mean of some groups are not part of a normal distribution (especially Control group of Run phase group, which probably include outliers). t-test does not apply to non-normal distribution. Normality test must be performed to check normality, otherwise authors must not rely on t-test to assume the statistical differences presented in Figure 6. In the legend of this same figure, authors say that Fisher's exact test was employed for promastigotes with migration speed higher than 5 μm /sec - this is called subsampling. Although subsampling is a valid approach, it must not be applied to Figure 6 results for the following reasons: a) Based on Figure 4, the number of promastigotes with speed higher than 5 μm /sec is extremely low or zero; conversely, the number of these promastigotes (>5 μm /sec) tested in Figure 6 does not represent the population of some groups, is a subsampling based on a minority of promastigotes with higher speed. b) To validate authors hypothesis (PI decrease parasite speed in COLI matrices), authors must look for differences in the entire population of promastigotes, not a subsample of the faster promastigotes. If (and only if) the whole population of promastigotes migrates slower in the presence of PI, than a subsampling may be performed in order to checki differences in the speed of the faster promastigotes. Otherwise, the Fisher's exact test in this subgroup (>5 μm /sec) doesn't add much.

5- The most interesting and creative part of this study is the addition of macrophages in the COLI matrix/promastigote system. This is extremely important and largely neglected by others who experimented promastigote invasion in ECM mimics. If parasites are able to invade macrophage-containing matrices (Figure 8) and there interact with these host cells (Figures 9 and 10) we must inquire: a) are these macrophages able to phagocytose promastigotes in this system? b) What about the infection indices or percentages in systems with or without PI? These questions could be answered by the observation and quantification of promastigotes (labeled with green cell tracker before addition to matrices) inside macrophages (fixed after 6h, 24h and 48h of promastigote addition and labeled with phaloidin).

Minor issues:

- The manuscript needs extensive and careful language and scientific revisions. I strongly recommend a proofreading service. Among several other instances I will cite the following:

- Line 22- "However" should be changed to "and" (there is no apparent relation of antagonism between phrases).

- Line 23- There is no beginning of a parasite life cycle, as in "Its life cycle begins". Authors should start the phrase in "The promastigote form of L. amazonensis is transmitted through the bite of infected sand flies from the genus Lutzomya (...). Once inoculated, ..."

- The last two paragraphs of Introduction fit best in Discussion. Additionally, it is quite inappropriate to reference figures in Introduction (Lines 56 and 64).

- In Methods, Line 86, what's the Rat Tail extracted COL I solution distributor? What's the substrate for COL I polymerization (Line 89), is it a tube? A 24 well plate? What's the live and dead kit employed (Line 92?)

- Abbreviation of "hour" must be normalized. It is abbreviated as hr (Line 89) and as h (Line 96) and it's not abbreviated in some instances. This must be checked in the entire manuscript.

- "Moved"(Line 108), "otherwise" (Line 131) and "retired"(Line 153) do not seem appropriate to me in the context of their phrases. "Modes"(Line 154) perhaps should be changed to models or ways.

- "Zimography" (Line 189) must be changed to Zymography, and o-fenantrolin must be changed to o-phenanthroline (Line 191).

- Figures and Videos must contain some arrowheads to point out events authors want to highlight. Reference bars (μm) and time bars (minutes and seconds) are missing in videos.

- In Discussion, Line 255, L. amazonensis is also able to multiply inside cells other than macrophages (fibroblasts are an excellent example). Line 258: promastigotes may contact other cells after interaction with ECM, such as neutrophils. Authors must check recent literature.

- "Mechanics" should not be regarded as merely physical or dynamic (Lines 270 and 277). The COL I matrices are physical (not mechanical) supports to promastigotes; these matrices could mimic the dynamics (not the mechanics) of human host early environment to Leishmania.

- "is important for infection" should be corrected to "are important for infection".

- There are excessive vague terms in the manuscript to interpret results. For example: Line 165, "strongly interact" is vague because authors did not show if the interaction in strong or weak (and in what sense), but only that they are interacting for sure; Line 283 "in several ways" means matrix degradation and migration, authors should be more precise.

---

## Round 0.2 · Minor Revisions

There is strong interest from the reviewers but some minor changes are necessary.

Reviewer 1 ·

Basic reporting

The authors present data on the establishment of L. amazonensis promastigote culture in a 3D environment mainly composed by collagen I to study L. amazonensis-extracellular matrix relationship.

Some suggested corrections were not complied:

The three first paragraphs of the introduction could be deleted (Leishmaniasis… Neuber 2008. Line 18 to 25). This information is very ordinary. Introduction could get straight on the subject of the manuscript;

Insert the reference “Silva-Almeida et al. 2012a” together with Abreu-Silva et al. 2004 and Melo et al. 2009. Line 31);

Supplementary materials were suppressed?

Experimental design

Experimental design is OK.

Validity of the findings

The findings are very interesting and the data from the basis are important to contribute to the Leishmania-extracellular matrix interaction knowledge.

Additional comments

Some suggested corrections were not complied.

Reviewer 2 ·

Basic reporting

See General Comments for the Author

Experimental design

See General Comments for the Author

Validity of the findings

See General Comments for the Author

Additional comments

Leishmania amazonensis promastigotes in 3D Collagen I culture: an in vitro physiological environment for the study of extracellular matrix and cell host interactions.

Review: R1 version

(Line numbers from MANUSCRIT_RESUBMITED_2_FIXED_REFS.DOCX file).

This reviewer appreciated the extensive and careful revision of the manuscript (which improved a lot) and also the convincing rebuttal of all major concerns raised. I should insist, however, on the discussion of some aspects that I consider relevant for a scientifically (not only methodologically) accurate and sound publication in PeerJ.

1- Concerning infective vs non-infective parasites on the degradation of collagen matrix: I disagree that testing non-infective promastigotes (procyclics) is out of the scope of the manuscript. The authors declared in their rebuttal: "We choose to use stationary phase promastigotes (mainly metacyclics) because they are known to be the relevant form during vector inoculation in the mammalian host dermis (Teixeira et al., 2013; El-Hani et al., 2012; Warderley et al., 2009)." Metacyclic forms do not mainly compose stationary populations: depending on strain and species, less than 30% of stationary promastigotes are in fact metacyclics. This seems to be the case of L. amazonensis (see Saraiva et al., 2005 and also Wanderley et al., 2009, which clearly show distinct populations of promastigotes inside stationary phase cultures). So, in the cultivation system employed by the authors, the majority of promastigotes is probably not composed by metacyclics (the infective, relevant forms to establish infection, so far recognized). In other words, the majority of promastigotes employed in the manuscript, although belonging to stationary phase cultures, is probably non-infective. Thus, a scientifically crucial question remains: Is COL I degradation important for infection or is just an artifact of Leishmania-COL I matrix co-cultivation? The authors should at least discuss this point more extensively in their manuscript.

2- 3D terminologies and jargons: this reviewer appreciated the references on 2D/3D culture models and is aware that this terminology has been used for 30 years or more. Of course it can be employed in the manuscript, but authors must keep in mind that PeerJ is a journal for a broad audience and that this terminology (2D/3D culture model) is a jargon of tissue/cell-culture, ECM specialists. If used without parsimony or accuracy, this terminology can lead to misconceptions, redundancies and conflict concerning spatial/dimensional terminologies. For example:

- The term "3D environment" (lines 319-320) is conceptually redundant, no matter the context (parasitology, cell culture, etc). In my opinion, it should be changed to "environment of the 3D culture", considering (in the introduction) that "3D culture" is presented as a system/type of cultivating cells classified in contrast to what is conventionally called "2D culture". Three-dimensionality (or 3D) must not be used as a synonym of "3D culture".

- "3D in vitro cultivation" (lines 290-291) could be changed to "in vitro cultivation in the 3D matrices (or cultures)". "3D interaction model" (lines 246-265) could be changed to "interaction model in 3D culture" or "interaction model in the 3D COL I matrix".

- The heading "Promastigote-macrophage 3D interaction" (line 253) is redundant since host cell-parasite interactions are three-dimensional phenomena. It should be changed to "Interaction of macrophages and promastigotes in 3D COL I matrix" which exactly depicts what is presented in the results.

- I insist that authors should not use the term "3D migration" (line 250, lines 306-307) when presenting their results of promastigote migration and transmatrix migration. It should be changed to "migration in 3D cultures", which has completely different meanings. As previously stated by this reviewer, a 3D migration data integrate x,y and z coordinates in the analysis of promastigote displacements (not performed). What authors did was evaluate x, y (migration) and x, z (transmatrix migration) separately, thus presenting results of promastigote migration in 3D COL I matrices.

These are examples of the kind of accuracy in 3D terminology I was expecting from the authors.

3- t-tests and the statistics of Figure 6: the authors are right to use Mann-Whitney test in this figure, which assumes non-normal distributions - this test is more reliable for the results presented in the figure 6. Authors must consider an important feature of their population: the small number of analyzed promastigotes (15 per videomicrography, 3 videomicrographies acquired at 24, 48 and 72 hours per condition, if I understood correctly). When sampling promastigotes in freeze-run and run groups, this reviewer assumes that authors are analyzing samples with n<50 in some conditions. It is not reliable to use t-test for such small non-normal samples, especially when we cannot predict if they rely on the Central Limit Theorem (CLT).
Lumley et al, 2002 (referenced in author's rebuttal) gives us a lot of evidence that t-test can in fact be employed in non-normal distributions, provided they belong to large samples and/or rely on CLT. This is not the case of figure 6: n<50 in all or some groups and we cannot predict that they will fit a normal distribution if n is increased (thus relying on CLT). Authors must include in the legend of figure 6 the n (cases analyzed) per group and the results of the Mann-Whitney test.

Other minor points:

- Title: The new proposed title is far better than the previous one. However, I suggest a minor modification in “cell host”: “Leishmania amazonensis promastigotes in 3D Collagen I culture: an in vitro physiological environment for the study of extracellular matrix and host cell interactions.”

- Lines 27-28: Change to "Because L. amazonensis is an intracellular parasite that only proliferates inside a host cell in the mammalian host". It proliferates in the extracellular milieu of insect hosts.

- Line 37: "The glycoprotein gp63 present on parasite surface is a broad-spectrum...". This phrase clarifies that gp63 is a parasite glycoprotein.

- Line 80-81: There are several Live/Dead fluorescent kits from Invitrogen (http://www.lifetechnologies.com/br/en/home/brands/molecular-probes/key-molecular-probes-products/live-dead-viability-brand-page.html). Which one was used? Please, include this information in methods.

- Line 130-131: "The degradation assay was adapted from Sugiama et al., 1980".

- Line 174-177: I insist on recommending the avoidance of vague terms such as huge, extensively and dramatically. This is not "intuitively understandable terminology", it is unnecessary and vague, not scientific. "Huge" can be interpreted as billions, millions, hundreds. "Extensively" is also vague and does not depicts properly what authors are describing in the figures. May I suggest, to improve concision "Scanning Electron Microscopy (SEM) of promastigotes cultivated in a 3D COL I matrix revealed that promastigotes adhered to collagen fibers (Fig. 1). The presence of promastigotes altered the organization of the COL I fibers network, from a homogeneous pattern (Fig. 1A) into a meshwork divided in areas of high fiber density and large fiber-free channels (Fig. 1E, F)."?

- Line 210-211: "The migration speed in 3D cultures significantly...".

- Line 265: "a real barrier" should be changed to "an effective barrier".

- Lines 268-271: This part of the text needs to be rewritten and clarified. "The injection of promastigotes directly into the matrix containing macrophages allowed for a good distribution of promastigotes inside the collagen meshwork and the observation of macrophage-promastigote interaction in the first minutes of interaction.". Additionally: why is it curious that promastigotes migrated faster and further than macrophages in the COL I system? Should the macrophages be faster than Leishmania promastigotes? Do authors refer to transmatrix migration? These lines are cryptic to me.

- Lines 275-276: The statement that L. amazonensis only proliferates in macrophages in human hosts is an overstatement. "Mainly" instead of "only" could be more appropriate to use in the phrase.

- Line 279: "dermis ECM"

- Line 280: "...with macrophages and other potential host cells".

- Lines 305-307: This phrase should be rewritten. "Protease blockage by a mixture of inhibitors decreased matrix degradation and affected promastigote migration and transmigration among COL I fibers".

- Lines 341-343: This phrase is almost a repetition of lines 322-323. Perhaps the first phrases of this paragraph (line 341) should be deleted, beggining with "We showed that promastigotes degrade COL I matrix via ..."

·

Basic reporting

No comments.

Experimental design

No comments.

Validity of the findings

No comments

Additional comments

The authors changed or explained all raised questions and now the revised manuscript is acceptable for publication.

---

## Round 0.3 · accepted · Accept

Thank you for addressing all the concerns raised by the reviewers and making modifications accordingly.